# *Piriformospora indica* Increases Resistance to *Fusarium pseudograminearum* in Wheat by Inducing Phenylpropanoid Pathway

**DOI:** 10.3390/ijms24108797

**Published:** 2023-05-15

**Authors:** Liang Li, Ruiying Hao, Xiurong Yang, Yu Feng, Zhenghui Bi

**Affiliations:** 1School of Chemical Engineering and Technology, Hebei University of Technology, Tianjin 300401, China; haory123@126.com (R.H.);; 2Institute of Plant Protection, Tianjin Academy of Agricultural Sciences, Tianjin 300401, China

**Keywords:** *Fusarium pseudograminearum*, fusarium crown rot, *Piriformospora indica*, phenylpropanoid pathway

## Abstract

Fusarium crown rot (FCR), mainly caused by *Fusarium pseudograminearum*, not only seriously threatens the yield and quality of wheat, but also endangers the health and safety of humans and livestock. *Piriformospora indica* is a root endophytic fungus that colonizes plant roots extensively and can effectively promote plant growth and improve plant resistance to biotic and abiotic stresses. In this study, the mechanism of FCR resistance mediated by *P. indica* in wheat was revealed from the phenylpropanoid metabolic pathway. The results showed that the colonization of *P. indica* significantly reduced the progression of wheat disease, the amount of *F. pseudograminearum* colonization, and the content of deoxynivalenol (DON) in wheat roots. RNA-seq suggested that *P. indica* colonization could reduce the number of differentially expressed genes (DEGs) in the transcriptome caused by *F. pseudograminearum* infection. The DEGs induced by the colonization of *P. indica* were partially enriched in phenylpropanoid biosynthesis. Transcriptome sequencing and qPCR indicated that the colonization of *P. indica* up-regulated the expression of genes involved in the phenylpropanoid biosynthesis pathway. The metabolome analysis indicated that the colonization of *P. indica* increased the metabolites’ accumulation in the phenylpropanoid biosynthesis. Consistent with transcriptome and metabolomic analysis, microscopic observations showed enhanced lignin accumulation in the roots of the Piri and Piri+Fp lines, most likely contributing to the arrested infection by *F. pseudograminearum.* These results suggested that *P. indica* increased resistance to *F. pseudograminearum* in wheat by inducing the phenylpropanoid pathway.

## 1. Introduction

Fusarium crown rot (FCR) is a devastating fungal disease of wheat and other small grain cereals caused by *F. pseudograminearum.* From its first detection in Australia in 1951, FCR has been found in many other countries [1,2,3]. According to the investigation, *F. pseudograminearum* is becoming a predominant causative pathogen of crown rot of wheat in Eastern China [3]. FCR results in a 9–35% yield reduction every year in China and up to 70% in severely affected areas with an associated reduction in quality [4]. The biggest concern, however, is that the disease causes large amounts of mycotoxins in plants that are harmful to the health of humans and animals [5]. *F. pseudograminearum* produces a plethora of trichothecene mycotoxins, including deoxynivalenol (DON), its acetylated derivatives (3aDON, 15aDON), and nivalenol (NIV) [6]. DON has been shown to be the most common mycotoxin contaminant causing hematopathia and anorexia as well as neurotoxic and immunotoxic effects in mammals [7]. Several countries and governments have recently established legal limits for DON in wheat and/or cereals and their derivatives for human consumption. Currently, the WHO is discussing advisory levels for these mycotoxins [6].

Due to the limited genetic resources for host resistance and the wide range of FCR pathogens, the application of fungicides is the main tool for controlling FCR. However, the high selection pressure caused by the widespread use of fungicides has increased the development of fungicide resistant pathogens. The results showed that the DON content in carbendazm-resistant *F. graminearum* isolates was significantly higher than that of carbendazm-sensitive isolates [8], which may have consequences for food safety. Therefore, increasing crop yields should be performed in a way that is reliable for producers, safe for consumers, and friendly to the environment. Recently, the agricultural practice of using rhizosphere endophytes to develop biodefenses as a safer and more productive means to combat plant diseases has attracted more attention to researchers and scientists [9].

*Piriformospora indica* (*P. indica*) is a root-endophytic basidiomycete member of the order Sebacinales, which was first discovered in the Thar desert in northwestern India in 1998 [10]. *P. indica* is well known to be able to establish beneficial interactions with an exceptionally large plant host range, including monocotyledon plants such as barley, wheat, rice, corn, and dicotyledon plants such as Arabidopsis and tobacco [11]. *P. indica* is an excellent model of beneficial microorganisms because the fungus can be genetically transformed and cultured under axenic conditions [12]. Additionally, it can widely promote plant growth and yield [13]. It can also enhance plant resistance to biotic stress, including pathogenic fungi [14] and viruses [15], as well as abiotic stresses [16].

The disease resistance of *P. indica* endowed to plants has made it an increasing focus of scientific attention. It has been confirmed that *P. indica* can enhance plant resistance to *Cochliobolus sativus* [17] and can induce systemic resistance to powdery mildew in barley [18] and *Verticillium dahlia* in Arabidopsis [19]. *P. indica* colonization reprogrammed transcriptome and effectively activate jasmonic acid/ethylene (JA/ET)-mediated basic defense mechanisms against pathogen infection [20]. *P. indica* can trigger changes in hormone levels, including salicylic acid (SA), gibberellin (GA), and abscisic acid (ABA), leading to host-microbial-associated molecular model (MAMP) immunity [21] and symbiotic self-regulation [22,23]. Studies have shown that defensive secondary metabolites in plants, such as artemisinin, abricin, triterpenes, and curcumin are produced by the colonization of *P. indica* [24,25,26,27]. It was reported that the colonization of *P. indica* could improve plant resistance to *F. graminearum* by initiating phenylpropanoid biosynthesis [9]. Phenylpropanoid metabolism is one of the most important metabolic pathways in plants, producing more than 8000 metabolites that contribute to plant development and plant–environment interaction. In response to biotic and abiotic stresses, and to cope with positive and negative ecological factors, plants have evolved elaborate biosynthetic mechanisms for secondary metabolites such as terpenoids, phenylpropyl, and alkaloids [28]. One of the products of the phenylpropanoid metabolic pathway is flavonoid, which plays an important role in promoting plant growth and development, resisting biotic and abiotic stresses [29]. Flavonoids are one of the most active antioxidants in plant stress resistance, which can regulate redox system factors; activate plant immune system; trigger the production of peroxidase (POD), superoxide dismutase (SOD), catalase (CAT), and other enzymes; and participate in plant disease resistance [30]. Therefore, secondary metabolites, otherwise called specialized metabolites, play a key role in plant growth and development [31,32,33]. The role of phenylpropanoid in wheat colonized by *P. indica* against Fusarium has not been elucidated, and their participation in *P. indica*-induced signal transduction against Fusarium is unknown.

The aim of this study was to reveal the molecular mechanism of disease resistance to FCR induced by *P. indica* in wheat. We discovered that the precolonization of wheat roots by *P. indica* could alleviate root rot caused by *F. pseudograminearum*. Transcriptome data showed that 167 DEGs involved in phenylpropanoid biosynthesis pathway were regulated upon *P. indica* colonization. The precolonization of *P. indica* could trigger changes in the transcriptome and metabolome, and the transcriptome and metabolome were consistent in the metabolic pathway of phenylpropanoid. We speculated that the phenylpropanoid biosynthesis pathway participated in plant resistance to *F. graminearum* under *P. indica* colonization.

## 2. Results

### 2.1. P. indica Promote Root Growth and Plant Development

To investigate whether wheat seeds could develop beneficial interactions with *P. indica* chlamydospore suspensions, seedlings after five days after germination in soil were subjected to colonization identification. The hyphae of *P. indica* were widely distributed over the root surface (Figure 1A) that indicated that the establishment of the beneficial symbiosis was successful. SEM indicated that dense mycelia were generated by the germination of chlamydospores, which intertwined to form mycelial networks on the root surface (Figure 1C).

### 2.2. P. indica Induces Disease Resistance to F. pseudograminearum

In order to evaluate disease resistance to *F. pseudograminearum*, the wheat seeds pre-inoculated with *P. indica* were subjected to the infection. *F. pseudograminearum* infection led to severe stem browning and rot at the early stages of infection, but this phenomenon could be remitted by *P. indica* pre-colonization (Figure 2A). Disease index at different days including 5, 10, 15, 20, and 25 days were evaluated after the inoculation of *F. pseudograminearum*. With the prolongation of infection time, the disease resistance to *F. pseudograminearum* in the *P. indica* pre-colonized seeds gradually appeared (Figure 2B). The statistical data showed that plant and stem dry weights decreased significantly under the influence of *F. pseudograminearum* infection. However, the seedlings pre-colonized with *P. indica* were less affected by pathogen infection (Figure 2C,D).

### 2.3. Seeds Precolonized by P. indica Reduced the Colonization of Pathogen F. pseudograminearum

In order to analyze the reason for the reduction in disease index, the amount of *F. pseudograminearum* colonization in wheat roots was quantified by fluorescence quantitative PCR. The result showed that precolonization of *P. indica* in wheat roots significantly reduced the amount of pathogen colonization (Figure 3A). Furthermore, the content of DON toxin in wheat roots was detected in Fp and Piri + fp treatments. The DON toxin content in wheat roots increased with the accumulation of *F. pseudograminearum* infection time; however, the toxin content decreased in wheat roots that precolonized by *P. indica*. For example, after 20 days infection, the amount of DON in Fp treatment was 208.6 µg/kg, while the amount of DON in Piri + Fp was 178 µg/kg (Figure 3B).

### 2.4. Expression Profile Analysis in Wheat Responsive to Piriformospora indica and F. pseudograminearum Colonization

In order to study the effect of different treatments on the expression profile of wheat root, the transcriptomes of wheat responsive to *P. indica* and *F. pseudograminearum* colonization were obtained by the BGI Seq500 platform. Differentially expressed mRNAs (DEGs) in wheat were found in the different comparison groups, including Piri vs. Mock, Fp vs. Mock, Piri + Fp vs. Mock, Piri + Fp vs. Fp, and Piri + Fp vs. Piri (Appendix A). Analysis of unique DEGs in Fp-mock vs. Piri + Fp-mock revealed that 9762 DEGs were exclusively present in Fp-mock, 5923 DEGs were exclusively present in Piri + Fp-mock, and 21,593 DEGs were present in both comparison group (Figure 4A). For the reads from the putative Fp-mock vs. Piri-mock vs. Piri + Fp-mock, 9687 DEGs were exclusively present in the Fp-mock, 5885 DEGs were exclusively present in the Piri + Fp-mock, 382 DEGs were exclusively present in the Piri-mock, and 146 DEGs were present in the three comparison groups (Figure 4B). For the reads from the putative Fp-mock vs. Piri-mock vs. Piri + Fp-mock vs. Piri + Fp-Fp, 8252 DEGs were exclusively present in the Fp-mock, 4926 DEGs were exclusively present in the Piri + Fp-mock, 379 DEGs were exclusively present in the Piri-mock, 346 DEGs were exclusively present in the Piri +Fp-Fp, and 41 DEGs were present in the four comparison groups (Figure 4C). Through analysis and comparison of the above data, it can be seen that the addition of *P. indica* reduced the number of DEGs by 3839 in the Piri + Fp group in addition to the co-occupied differential genes (Figure 4A).

Concretely, when comparing the Fp vs. mock group, it was found that the number of down-regulated DEGs (17,828) was higher than the number of up-regulated DEGs (13,527) (Figure 5A). Among them, the DEGs were mainly enriched in photosynthesis antenna protein pathway according to the Kyoto Encyclopedia of Genes and Genomes (KEGG) pathway analysis (Figure 5B). Similarly, in the Piri vs. Mock group, the number of down-regulated DEGs (507) was higher than the number of up-regulated DEGs (134) (Figure 5C). The difference was that the number of DEGs decreased significantly and were mainly enriched in MAPK signaling pathway, plant–pathogen interaction, and phenylpropanoid biosynthesis (Figure 5D). A similar situation also occurred in the Piri + Fp vs. Mock comparison group, where the number of down-regulated genes (17,425) was higher than that of the up-regulated genes (10,091) (Figure 5E). KEGG analysis suggested that the DEGs were mainly enriched in the phenylpropanoid biosynthesis and the MAPK signaling pathway (Figure 5F). In the Piri + Fp vs. Fp group, however, the number of up-regulated genes (10,836) was higher than the number of down-regulated genes (9921) (Figure 5G), and these DEGs were mainly enriched in phenylpropanoid biosynthesis (Figure 5H). Additionally, it was evident that many DEGs were enriched in the phenylpropanoid biosynthesis pathway in the Piri vs. Mock, Piri + Fp vs. Mock, and Piri + Fp vs. Fp comparison groups.

### 2.5. Key DEGs Involved in Phenylpropanoid Biosynthesis Pathway Expression Identification by QPCR

In order to further explore the role of the phenylpropanoid biosynthesis pathway in wheat disease resistance to *F. pseudograminearum* mediated by *P. indica*, we quantitatively analyzed the expression of key genes involved in the phenylpropanoid biosynthesis pathway. Over the past few decades, a considerable number of studies revealed the complex mechanisms that control the biosynthesis of phenylpropanoid metabolites (Figure 6A). These studies provided a comprehensive understanding of phenylpropanoid metabolism and a theoretical basis for plant genetic improvement and production of metabolites beneficial to human health [34,35,36,37,38]. The phenylpropanoid synthesis pathway generally has different branches, among which the lignin pathway and the flavonoid pathway are two main branches. During the *P. indica* colonization, we noticed that plenty of key DEGs involved in lignin and flavonoid biosynthesis pathway were induced according to the transcriptome sequencing. The up-regulated induction of key DEGs involved in lignin and flavonoid synthesis pathway are shown in Table 1 and Table 2, respectively. Furthermore, the expression levels of key genes involved in the lignin and flavonoid synthesis pathway were detected by fluorescence quantitative PCR (Figure 7 and Figure 8). The qPCR data indicated the key genes involved in lignin synthesis, including *PAL*, *CAD*, *COMT*, *CSE*, *4CL*, and *CCOACMT*, were all upregulated after *P. indica* colonization (Figure 7A–F). Additionally, the other key genes involved in flavonoid synthesis, including *CHS*, *CHI*, *FOMT*, *IFR*, *ANR*, and *UG*T, were all upregulated after *P. indica* colonization (Figure 8A–F). We conclude that the colonization of *P. indica* induces the up-regulated expression of genes involved in the phenylpropanoid biosynthesis pathway.

### 2.6. Metabolic Profiles of Wheat Root in Response to P. indica Colonization

To further elucidate the mechanism of disease resistance mediated by the phenylpropanoid biosynthesis pathway in response to the colonization of *P. indica,* non-targeted metabolic analysis was performed on the mock, Piri, Fp, and Piri + Fp treatment roots at 20 days after pathogen infection (Figure 9). The metabolite profiling of the Piri, Fp, and Piri + Fp treatments showed different changes in contrast to mock including flavonoid, phenylpropanoid, indole alkaloid, amino acids, fatty acids, terpenoids, etc. In the Piri treatment, a total of 460 metabolites were detected in the wheat roots under Piri treatment. Among them, the levels of 352 metabolites increased and 108 metabolites decreased (Figure 10); the metabolites of flavone and flavonol biosynthesis, tropane piperdine and pyridine alkaloid, and phenylpropanoid biosynthesis were significantly upregulated (Figure 9A). In the Fp treatment, a total of 850 metabolites were identified, of which the contents of 292 metabolites increased, and 558 metabolites decreased; and the metabolites of porphyrin and chlorophyll metabolism, lysine biosynthesis, and cyanamino acid metabolism were significantly down regulated (Figure 9B). Whereas, in the Piri + Fp treatments, a total of 930 metabolites were identified, of which the contents of 639 increased, and 291 decreased, and the metabolites of the flavonoids, phenylpropanoids, and indole alkaloids were significant upregulated (Figure 9C). Particularly, we noticed that the addition of *P. indica* significantly increased the metabolite contents of the phenylpropanoid biosynthesis pathway (Figure 9A,C).

### 2.7. Lignin Staining of Wheat Roots among Different Treatments

Histochemical staining of lignin with basic fuchsin showed that *P. indica* colonization could increase the fluorescence intensity by 45% (Figure 11B) compared with the control condition (Figure 11A); after *F. pseudograminearum* infection, the fluorescence intensity decreased by 5% (Figure 11C) compared to the control, whereas the *P. indica* pre-colonization wheat infected *F. pseudograminearum* showed only a moderate decrease in lignin staining, decreased by 20% when compared to only *P. indica*. However, it was still higher than the mock conditions (Figure 11D,E). These results suggested that a suberized cell wall barrier was formed in the surface layer of wheat roots that had been precolonized with *P. indica* to resist the invasion of *F. pseudograminearum*.

## 3. Discussion

In recent years, soil-borne diseases of wheat have become an important limiting factor for high yield, high quality, and high efficiency. Therefore, it has become an urgent problem to strengthen the research of soil-borne disease occurrence rules and comprehensive control technologies. Wheat crown rot, known as the cancer of wheat, once infected with wheat, will cause tiller and ear number reduction, plant dwarf and weak, grain weight reduction, and then easy-to-form white ear, resulting in basically no yield. *F. pseudograminearum* and *Fusarium culmorum* are the most common and globally important causal agents of crown rot of grain cereals [39,40,41,42]. *Fusarium pseudograminis* is the most widespread and has been reported in all soil types in all grain-growing regions [43,44]. Therefore, it is particularly urgent and necessary to prevent and control the crop diseases caused by *F. pseudograminearum*. Here, the biological control ability of *P. indica* against wheat crown rot was explored in this research.

Many studies have shown that *P. indica* can widely colonize the roots of various plants and promote their growth and development [11,45,46]. The colonization pattern of *P. indica* is intercellular rather than intracellular [47], which is consistent with the results of our scanning electron microscopy. However, what we notice is that the mycelium formed by *P. indica* on the surface of the root acts as a protective membrane to protect the root of the plant against outside invasion (Figure 1C). Wheat precolonized with *P. indica* showed significant resistance to *F. pseudograminearum*, and the disease index statistics showed that the resistance effect was more significant with the increase in the time interval between *P. indica* colonization and pathogen infection (Figure 2B). At the same time, it was further observed that wheat precolonized with *P. indica* could alleviate the loss of plant and stem dry weight due to *F. pseudograminearum* infection. These results were in accordance with the previous research [9].

In this study, the amount of *F. pseudograminearum* colonization in plant roots that pre-colonized by *P. indica* was quantitatively analyzed. The results showed that the amount of *F. pseudograminearum* infection decreased by approx. 50% in wheat roots that were precolonized with *P. indica* (Figure 3A). Furthermore, DON levels were measured at different time points after *F. pseudograminearum* infection. Compared with the pathogen infection alone, the data showed that the DON content in wheat roots of precolonizing *P. indica* was significantly decreased (Figure 3B). DON has been shown to be the most common mycotoxin contaminant causing hematopathies and anorexia in mammals, as well as neurotoxic and immunotoxic effects [7]. Precolonization of *P. indica* can reduce the accumulation of toxins caused by pathogen infection in wheat roots, which is undoubtedly a promising application for agricultural production and food safety.

The transcriptome data showed that *P. indica* did not cause much differential gene expression (641) in the wheat transcriptome despite colonization by fungi in vitro (Figure 4B), whereas it did (31,355) for *F. pseudograminearum* (Figure 4A). Moreover, the Venn diagram also showed that the addition of *P. indica* evidently reduced the significant genetic variation in the wheat transcriptome caused by *F. pseudograminearum* invasion (Figure 4A). At the same time, the common DEGs and the unique DEGs in each comparison group suggested that the vast functional resources of the genes remained to be explored (Figure 4C) for unrevealing the mechanism of *P. indica*-mediated FCR resistance. The enrichment analysis of the DEGs showed that some of the DEGs were enriched in the photosynthesis antenna protein pathway, MAPK signaling pathway, plant pathogen interaction and phenylpropanoid pathway, etc., among the different comparison group (Figure 5A–H). Notably, as long as there is precolonization of *P. indica*, there will be DEGs enriched in the phenylpropanoid metabolic pathway. Therefore, it is reasonable to speculate that the phenylpropanoid pathway plays an important role in wheat FCR resistance mediated by *P. indica.*

The phenylpropanoid metabolic pathway is one of the most important metabolic pathways in plants, which can produce more than 8000 metabolites and participate in plant development and plant–environment interactions [28]. Over the past few decades, numerous studies have revealed the complex mechanisms that control and regulate the biosynthesis of phenylpropanoid metabolites (Figure 6). Branches of phenylpropanoid metabolism yield end products such as flavonoids, hydroxycinnamate esters, hydroxycinnamate amides (HCAAs), and precursors to lignin, lignans, and tannins [48]. Homeostasis between the different branches of phenylpropanoid metabolism is achieved through the regulation of metabolic flux redirection (MFR), showing exceptional complexity and high levels of plasticity in successive developmental stages and in response to environmental stimuli and changes [49]. In our effort, we found the key genes in the synthetic pathway involved in the lignin and flavonoids biosynthesis were all up-regulated upon *P. indica* colonization and *P. indica* + *F. pseudograminearum* co-infection (Figure 7 and Figure 8). These results corroborated the reliability of the RNAseq results. Flavonoids, as important secondary metabolites, confer resistance to biotic and abiotic stress in plants. Xu et al. [50] reported that the flavonoids synthesis pathway was upregulated following powdery mildew infection and further supported the hypothesis that flavonoids play a key role in powdery mildew resistance in wheat.

To explore the disease resistance to *F. pseudograminearum* in wheat under colonization by *P. indica*, non-targeted metabolic analysis was performed on wheat roots. Different metabolic profiles were identified in wheat roots under different treatments (Figure 9A–C). Primary metabolic processes, including amino acid and lipid metabolism, were up-regulated under *P. indica* colonization when compared with the control plants. It is well known that plant growth and development are affected by the primary metabolic processes involved in the synthesis of proteins, carbohydrates, and lipid-related substances [13]. This is the reason why the colonization of *P. indica* can promote plant growth. Especially, the flavone and flavonol biosynthesis; phenylpropanoid biosynthesis; and tropane, piperidine, and pyridine alkaloid biosynthesis were significantly upregulated upon *P. indica* colonization (Figure 9A). In the treatment of *F. pseudograminearum* infection, more metabolites involved in lysine biosynthesis, cyanoamino acid, monobactam biosynthesis, and glycerophospholipid metabolism were found to be down-regulated (Figure 9B). Whereas, in the Piri + Fp treatment, the metabolites involved in flavonoids biosynthesis, indole alkaloid biosynthesis, and phenylpropanoid biosynthesis were upregulated (Figure 9C). Evidently, metabolome data showed that metabolite accumulations belonging to the phenylpropanoid metabolic pathway were induced in wheat resistance to *F. pseudograminearum* mediated by *P. indica*, which also confirmed the transcriptome results. Metabolome data further showed that the number of up-regulated metabolites in roots was lower than that of down-regulated metabolites after *F. pseudograminearum* treatment, while the number of up-regulated metabolites was greater than that of the down-regulated metabolites in both the Piri treatment group and the Piri + Fp group. This also suggests that metabolome expression profile could be modified upon *P. indica* colonization.

Finally, lignin, the ultimate metabolite of the phenylpropanoid pathway, was detected by histochemical staining. The lignin histochemical staining at the root surface cell layers was considerably enhanced in wheat lines after *P. indica* colonization (Figure 11B) compared to the mock (Figure 11A). However, the fluorescence intensity of lignin decreased in wheat roots infected by *F. pseudograminearum* (Figure 11C). On the other hand, the Piri + Fp lines showed only a moderate decrease in lignin staining compared to the *P. indica* treatment (Figure 11D), and still a 16% higher in fluorescence intensity than the mock conditions (Figure 11E). These results implied that the root surface colonized by *P. indica* formed a cell wall barrier, as shown in SEM (Figure 1C), to resist the invasion of *F. pseudograminearum.* The study [40] showed that suberin accumulation in roots of barley (HvMPK3) contributing to the arrested infection by *F. graminearum,* which was more or less consistent with our microscopic observations.

In conclusion, physiological data showed that colonization of *P. indica* significantly reduced the progression of wheat disease caused by *F. pseudograminearum* infection. The colonization of *P. indica* in wheat significantly decreased the colonization of *F. pseudograminearum* and the content of deoxynimofusciol (DON) in wheat roots. RNA-seq suggested that *F. pseudograminearum* led to serious reprogramming of the transcriptome, and *P. indica* colonization could reduce the number of DEGs of the transcriptome caused by the pathogen infection. The DEGs induced by the colonization of *P. indica* were partially enriched in phenylpropanoid biosynthesis. Transcriptome sequencing and QPCR indicated that the colonization of *P. indica* induced the up-regulated expression of genes involved in the phenylpropanoid biosynthesis pathway. Metabolome analysis indicated that the colonization of *P. indica* increased the metabolite accumulations in the phenylpropanoid biosynthesis. Consistent with transcriptome and metabolomic analysis, microscopic observations showed enhanced lignin accumulation in roots of Piri and Piri + Fp lines, most likely contributing to the arrested infection by *F. pseudograminearum.* Our study provided a fundamental theory for further exploration of the potential of phenylpropanoid biosynthesis as a biological prevention and control pathway against FCR in wheat.

## 4. Materials and Methods

### 4.1. Fungal Inoculums Preparation and Inoculation

*Piriformospora indica* (CGMCC No.10325) was provided by Karl-Heinz Kogel at the IPAZ, Giessen University, Germany. *F. pseudograminearum* isolated was provided by Tianjin Academy of Agricultural Sciences, China. The wheat seeds (Zhongmai 100) were provided by the Institute of Plant Protection, Chinese Academy of Agricultural Sciences, Beijing, China. *P. indica* growing on CM medium plates for 3–4 weeks was used to collect the spore suspensions. The detailed procedures were referenced by the literature [51]. The wheat seeds were surface sterilized for 20 min with 3% active chlorine and sodium hypochlorite solution and washed five times. Then, the seeds were soaked in chlamydospore suspensions of *P. indica* (1 × 10^5^/mL) for 30 min. Control seeds were treated with Tween 20 (0.002%, *v*/*v*) solution as mock. The mock and soaking seeds were sown at the same time, with six replicates for each treatment. Wheat biomass analyses were performed on seedlings grown on soil under 8 h dark (18 °C) and 16 h light (160 μmol m^−2^ s^−1^, 22 °C) conditions at 65% relative humidity for 2 weeks. Conidia of *F. pseudograminearum* growing on potato dextrose agar (PDA) media were harvested as described previously [52] and used for seedling infection. The wheat stem base was inoculated with 10 µL spore suspension (1 × 10^5^ macroconidia mL^−1^), using a syringe with an automatic dispenser. Inoculated plants were covered with moist plastic bags to maintain a saturated atmosphere and facilitate infection, and the bags were removed 48 h after inoculation.

### 4.2. Colonization Observation of P. indica

The *P. indica* colonization in wheat roots was identified by staining the root fragments using trypan blue according to the method of Li et al. [53]. Root samples were then subjected to microscopic observation using a Nikon ECLIPSE NI-U research microscope (Nikon, Tokyo, Japan). For Scanning Electron Microscopy (SEM) observation, the roots of the co-cultured seedlings were cut into small pieces, washed three times in double distilled water, and then soaked in 2.5% glutaraldehyde for 30 min. After several rinses with double distilled water, they were dehydrated in 50, 70, 80, 90, and 100% ethanol step-by-step (15 min each step). The samples were dried overnight at 4 °C, coated with ETD 2000C ion coater (Beijing Fine Technology Development Co., LTD., Beijing, China), and observed under ZEISS MERLIN compact scanning electron microscope (ZEISS, Jena, Germany).

### 4.3. Plant Disease Evaluation

Basal rot severity disease caused by *F. pseudograminearum* was graded into ten classes based on the proportion of stem discoloration (1 = 10%; 2 = 20%; 3 = 30%; 4 = 40%; 5 = 50%; 6 = 60%; 7 = 70% 8 = 80%; 9 = 90%; 10 = dead plant), according to the description by Fernandez and Chen [54] and modified some details.

### 4.4. Deoxynivalenol (DON) Assay

DON content was detected by ELISA kit provided by (Fuda Testing Group Co., Ltd., Shanghai, China). Samples, standard products, and HRP-labeled competing antigens were added into the coated micropores pre-coated with DON antigens, incubated, and washed thoroughly. Color was produced with the substrate TMB, which was converted to blue by the catalysis of peroxidase and to the final yellow by the action of acid. The shade of the color was negatively correlated with the DON in the sample. The absorbance (OD value) was measured at 450 nm wavelength, and the sample concentration was calculated. The minimum detected concentration was less than 10 PPb.

### 4.5. Quantification of F. pseudograminearum Colonization by QPCR

*F. pseudograminearum* TUBULIN (FpTUB) were determined via the −ΔΔ^Ct^ method by relating the amount of target transcript to the amount of Ta UBIQUITIN. Genomic DNA was isolated from 100 mg root tissue by using the DNeasy plant Mini Kit (Qiagen, Shanghai, Germany. http://www.qiagen.com.http.hebutlib.proxy.hebut.edu.cn, accessed on 8 October 2022), according to the manufacturer’s instructions. For quantitative real-time PCR, 10 ng of total DNA were used. Amplifications were performed in 15 μL of SYBR green JumpStart Taq ReadyMix with 0.5 pmol oligonucleotides, using Light Cycler96 Fast real-time PCR system (Roche, Basel, Switzerland). After an initial activation step at 94 °C for 5 min, 35 cycles (95 °C for 35 s, 58 °C for 30 s, 72 °C for 35 s, and 65 °C for 20 s) were performed. Respective melting curves were determined at the end of each cycle to ensure amplification of only one PCR product. Ct values were determined with the Cycler96 Fast software (SW1.1) supplied with the instrument. The primers used for all analyses are listed in Appendix A.

### 4.6. Histochemical Detection of Lignin

Root segments (first 25 mm from the root apex) of different treatment 48 h after inoculation with *F. pseudograminearum* were used for lignin histochemical detection. Lignin staining was performed according to the literature [55,56,57]. Stained cross-sections of roots were observed under confocal laser scanning microscope LSM710 (Carl Zeiss, Oberkochen, Germany) using following settings: 561 nm excitation wavelength, 0.6% laser power, the default gain, 600–650 nm emission bandwidth. Fluorescence intensity measurement and post-processing were performed using ZEN 2014 software (Carl Zeiss). Average fluorescence intensity was calculated from at least two wheat roots in three biological replicates (*n* = 6).

### 4.7. mRNA Library Construction

The mRNA library was constructed using the samples in Table 3. mRNA was purified by Oligo (dT) magnetic beads. The purified mRNA was split into small fragments with fragment buffer at appropriate temperature. Then, the first strand complement DNA (cDNA) was generated by reverse transcription using random hexamer primers, followed by the synthesis of the second strand cDNA. Terminal repair was then performed by incubation with the addition of a-tail mix and an RNA index adapter.

The cDNA fragments obtained in the previous step were amplified by PCR, purified with Ampure XP Beads, and dissolved in EB solution. The product was quality-control validated on the Agilent Technologies 2100 Bioanalyzer. The double-stranded PCR products of the previous step were heated and denatured by splinted oligonucleotide sequences and cycled to obtain the final library. DNA nanospheres (DNB) containing more than 300 copies of one molecule were amplified by phi29. DNB was loaded into a pattern nanoarray, and 100-base reads were generated on the bgisisq500 platform (BGI, Shenzhen, China).

### 4.8. GO and KEGG Enrichment Analysis of Differentially Expressed Genes

Cluster Profiler R package was used to enrich the gene ontology (GO) of differentially expressed genes and correct the gene length deviation. GO item with *p* < 0.05 was considered to be significantly enriched in differentially expressed genes. The Kyoto Encyclopedia of Genes and Genomes (KEGG) database (http://www.Thegenome.jp/kegg/ accessed on 15 May 2020) was applied for understanding the advanced functions and utility of biological systems, such as cells, organisms, and ecosystems, from information at the molecular level, particularly large-scale molecular data sets generated by high-throughput sequencing. We tested statistical enrichment of differentially expressed genes using cluster Profiler R packages and KOBAS 3.0 software. After correction, items with *p* values < 0.05 were considered to be significantly enriched in differentially expressed genes.

### 4.9. Quantitative Real Time Polymerase Chain Reaction (qRT-PCR) for Validation of Sequencing Results

The expression of each gene was determined by quantitative RT-PCR. Quantitative RT-PCR was performed by SYBR Green fluorescence method. In short, the qPCR experiment was performed on the Light Cycler96 Fast real-time PCR system (Roche). The reaction solution contained 2 × Ultra SYBR mix 10 µL, 100 ng cDNA template, and 10 µM forward and reverse primers. HvUBIQUTIN was used as the control, and all experiments were repeated at least three times. The amplification procedure was as follows: initial activation was performed at 95 °C for 5 min, followed by 30 cycles (95 °C, 20 s, 56 °C, 35 s, 72 °C, 65 °C, 20 s). At the end of each cycle, the melting curve was measured separately to ensure the amplification of a single PCR product. Primers used in the qPCR were listed in Appendix A.

### 4.10. Metabolic Analysis

At the jointing stage, ten roots were extracted from each treated plant as biological replicates, and 6 replicates were used for metabonomic analysis. Root samples from each group were cut into small pieces and extracted in a mixture of 1mL methanol and water (1:1, *v*/*v*). After full grinding, the extracted metabolites (5 μL) were loaded on Waters ACQUITY UPLC BEH C18 column (5 μm, 150 × 4.6 mm; water crops). Then, using the Waters UPLC I-Class/Xevo G2 xs—Quadrupole time-of-flight System (Waters Corp., Milford, MA, USA) at a flow rate of 0.3 mL/min at 4 °C, with liquid eluting with water (0.1% formic acid) and acetonitrile (0.1% formic acid). The detection conditions of mass spectrum were as follows: capillary voltage: 0.5 kV; Cone voltage: 35 V; Extractor voltage: 4.0 V; Source temperature: 11 °C; Desolubilization temperature: 550 °C. ESI-MS positive test. Metabolic data were collected and analyzed using MassLynx NT 4.1 software and the QuanLynx program (Waters Corp.). The relative concentrations (g/DW) of different metabolites in root tissues in each biological replicate were expressed as comma-separated values (.csv). Significantly regulated metabolites were identified in roots between the two groups (*P. indica* colonization was higher than that of the control), based on folding changes and error finding rates (Log2FC > 2 or Log2FC < 0.5, *p* < 0.05). Pathway analysis was performed using KEGG database (http://www.genome.ad.jp/kegg/pathway.html accessed on 27 July 2020) to better elucidate the functions of altered metabolites (*p* < 0.05 significant) in the clusterProfile package [58].

### 4.11. Statistical Analysis

In this study, all data were expressed as means ± SE and represented at least three independent biological experiments. One-way analysis of variance (ANOVA) with Duncan’s multiple range test method was used to analyze the significance of differences.

## Figures and Tables

**Figure 1 ijms-24-08797-f001:**
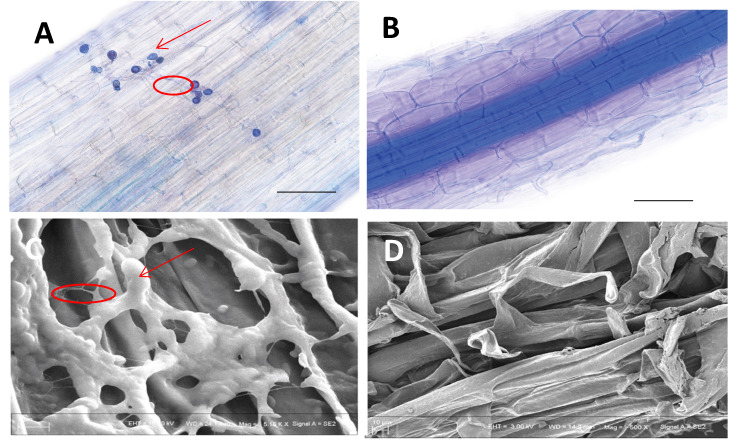
*P. indica* colonization identification in wheat roots. (**A**) *P. indica* colonizing roots form chlamydospores and germinates hypha on the surface of root cells. (**B**) Roots of non-inoculated wheat were used as the control. Fungal structures were visualized following staining with 0.05% trypan blue. (**C**) Scanning Electron Microscopy (SEM) images of the colonization process of *P. indica* in wheat roots. (**D**) SEM images of non-inoculated wheat as the control. Arrows indicates spores and circles indicated mycelia of *P. indica*.

**Figure 2 ijms-24-08797-f002:**
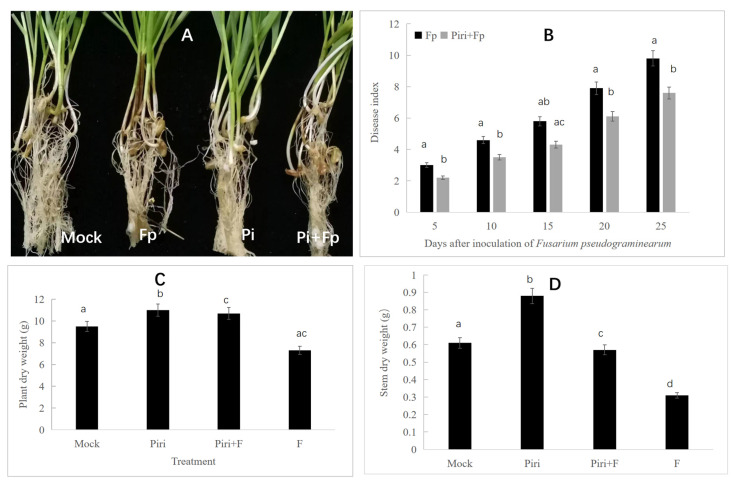
Disease resistance effect and growth parameter statistics of wheat to *F. pseudograminearum* mediated by *P. indica.* (**A**) Disease resistance effect to *F. pseudograminearum* mediated by *P. indica*. (**B**) The wheat seedlings were infected by *F. pseudograminearum* at 14 days after inoculation (day) with *P. indica* and harvested at different time-points, including 5, 10, 15, and 25 days. Effects of different treatments, including Fp and Piri + Fp, on progress of the disease on the wheat seedlings were evaluated. (**C**) Plant dry weight was calculated under different treatment including Mock, Piri, Fp, and Piri + Fp. (**D**) Stem dry weight was determined under different treatment including Mock, Piri, Fp, and Piri + Fp. Lowercase letter a, b, c, d indicated significance.

**Figure 3 ijms-24-08797-f003:**
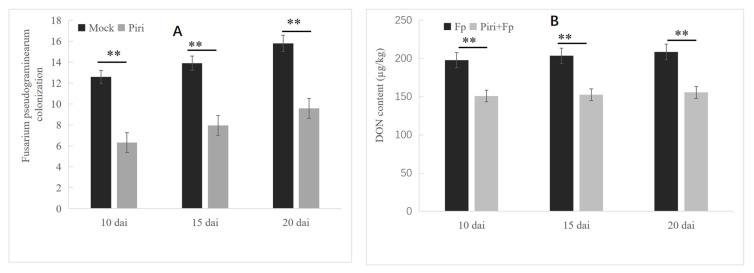
Quantification of *F. pseudograminearum* colonization and DON content in roots of wheat under different treatments. (**A**) Quantification of *F. pseudograminearum* colonization in Mock and *P. indica* pre-colonization roots. Mock and *P. indica* pre-colonization roots infected by *F. pseudograminearum* were harvested at day 10, day 15, and day 20 and used for *F. pseudograminearum* colonization analysis. (**B**) DON content in roots infected by *F. pseudograminearum* and *P. indica* + *F. pseudograminearum* were identified. Mock and *P. indica* pre-colonization roots infected by *F. pseudograminearum* were harvested at day 10, day 15, and day 20 and used for DON content quantification. ** above the columns represents significant differences (*p* < 0.01).

**Figure 4 ijms-24-08797-f004:**
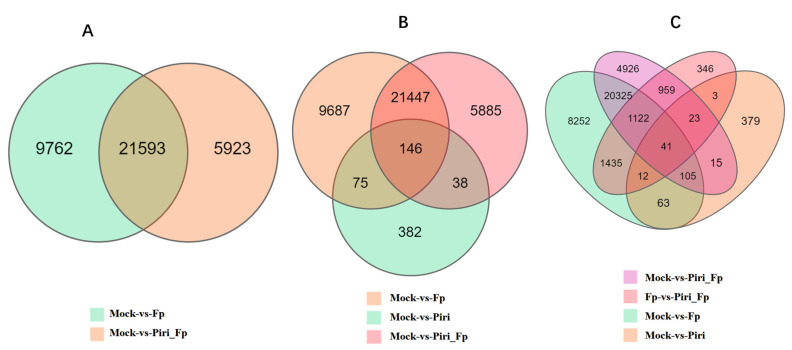
Venn diagrams showing the sample-exclusive or communal presence of unique putative endogenous DEGs. (**A**) putative endogenous DEGs in Mock-Fp vs. Mock-Piri_Fp; (**B**) putative endogenous DEGs in Mock-Fp vs. Mock-Piri vs. Mock-Piri_Fp; (**C**) putative endogenous DEGs in Mock-Piri_Fp vs. Fg-Piri_Fg vs. Mock-Fp vs. Mock-Piri.

**Figure 5 ijms-24-08797-f005:**
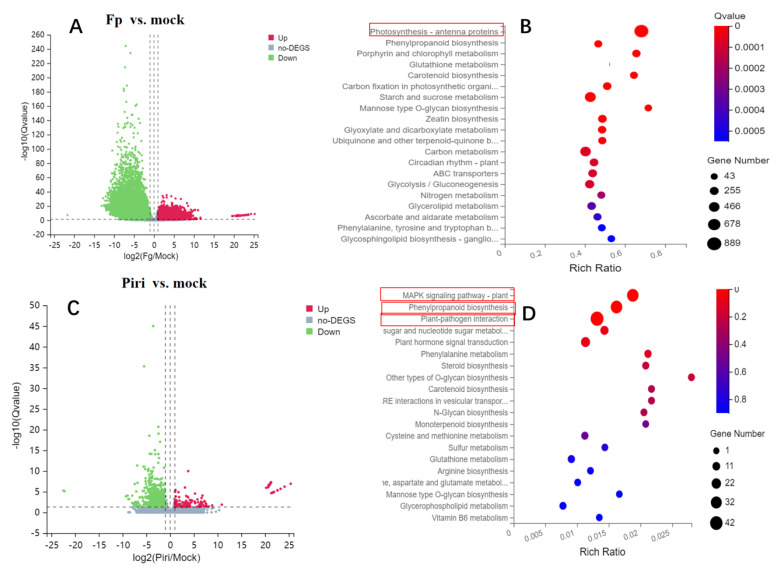
Profile of DEGs in wheat responsive to *P. indica* and *F. pseudograminearum* colonization. (**A**) Volcano plot of DEGs in Fp vs. Mock group; (**B**) DEGs enriched in KEEG pathway in Fg vs. Mock group; (**C**) Volcano plot of DEGs in Piri vs. Mock group; (**D**) DEGs enriched in KEEG pathway in Piri vs. Mock group; (**E**) Volcano plot of DEGs in Piri + Fp vs. Mock; (**F**) DEGs enriched in KEEG pathway in Piri + Fp vs. Mock; (**G**) Volcano plot of DEGs in Piri + Fp vs. Fp; (**H**) DEGs enriched in KEEG pathway in Piri + Fp vs. Fp. (/log2 FC/≥ 1, Q-value ≤ 0.05). Piri, *P. indica*; Fp, *F. pseudograminearum*; Piri + Fp, *P. indica* + *F. pseudograminearum*.

**Figure 6 ijms-24-08797-f006:**
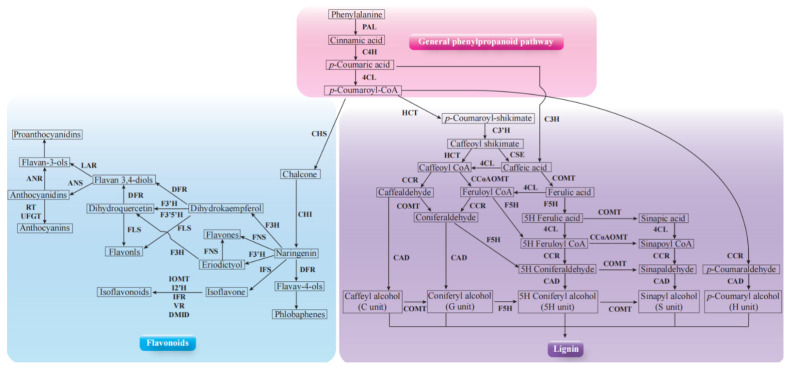
A scheme of phenylpropanoid metabolism in plants. Bold font indicates enzymes. 4CL, 4-coumarate-CoA ligase; ANR, anthocyanidin reductase; ANS, anthocyanin synthase; C3H, coumarate 3-hydroxylase; C3′H, p-coumaroyl shikimate 3′ hydroxylase; C4H, cinnamic acid 4-hydroxylase; CAD, cinnamyl alcohol dehydrogenase; CCoAOMT, caffeoyl CoA3-O-methyltransferase; CCR, cinnamoyl-CoA reductase; CHI, chalcone isomerase; CHS, chalcone synthase; COMT, caffeate/5-hydroxyferulate3-O-methyltransferase; CSE, caffeoyl shikimate esterase; DFR, fihydroflavonol 4-reductase; DMID, 7, 2′-dihydroxy, 4′-methoxyisoflavanol dehydratase; F3′5′H, flavonoid 3′5′-hydroxylase; F3H, flavanone 3-hydroxylase; F3′H, flavonoid 3′-hydroxylase; F5H, ferulate 5-hydroxylase; FLS, flavonol synthase; FNS, flavone synthase; HCT, Hydroxycinnamoyl-CoA shikimate/quinate hydroxycinnamoyl transferase; I2′H, isoflavone 2′-hydroxylase; IFR, isoflavone reductase; IFS, isoflavone synthase; IOMT, isoflavone O-methyltransferase; LAR, leucoanthocyanidin reductase; PAL, phenylalanine ammonialyase; RT, rhamnosyl transferase; UFGT, UDPG-flavonoid glucosyltransferase; VR, vestitone reductase.

**Figure 7 ijms-24-08797-f007:**
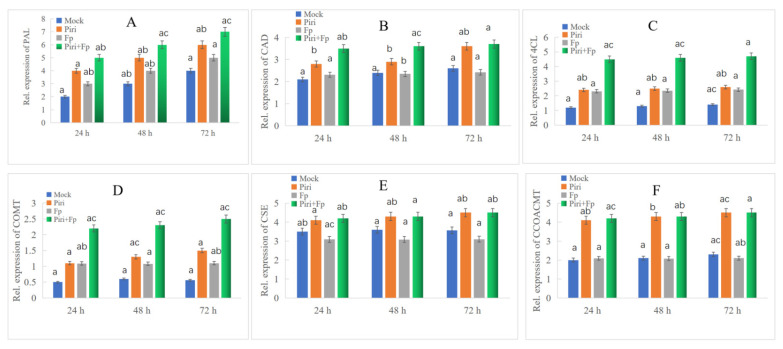
QPCR identification of the key genes participated in lignin biosynthesis pathway. The key genes involved in lignin biosynthesis, including (**A**) PAL, phenylalanine ammonialyase; (**B**) CAD, cinnamyl alcohol dehydrogenase; (**C**) 4CL, 4-coumarate-CoA ligase; (**D**) COMT, caffeate/5-hydroxyferulate3-O-methyltransferase; (**E**) CSE, caffeoyl shikimate esterase; (**F**) CCoAOMT, caffeoyl CoA3-O-methyltransferase; were all upregulated in Piri and Piri + Fp treatments. Different small letters above the columns represent significant differences (*p* < 0.05).

**Figure 8 ijms-24-08797-f008:**
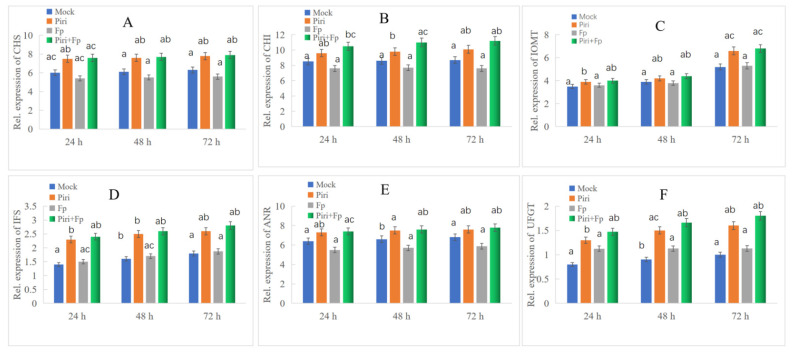
QPCR identification of the key genes participated in flavonoids biosynthesis pathway. (**A**) Key genes involved in flavonoids biosynthesis, including CHS, chalcone synthas; (**B**) CHI, chalcone isomerase; (**C**) IOMT, isoflavone O-methyltransferase; (**D**) IFS, isoflavone synthase; (**E**) ANR, anthocyanidin reductase; (**F**) UFGT, UDPG-flavonoid glucosyltransferase was all upregulated in Piri and Piri + Fp treatments. Different small letters above the columns represent significant differences (*p* < 0.05).

**Figure 9 ijms-24-08797-f009:**
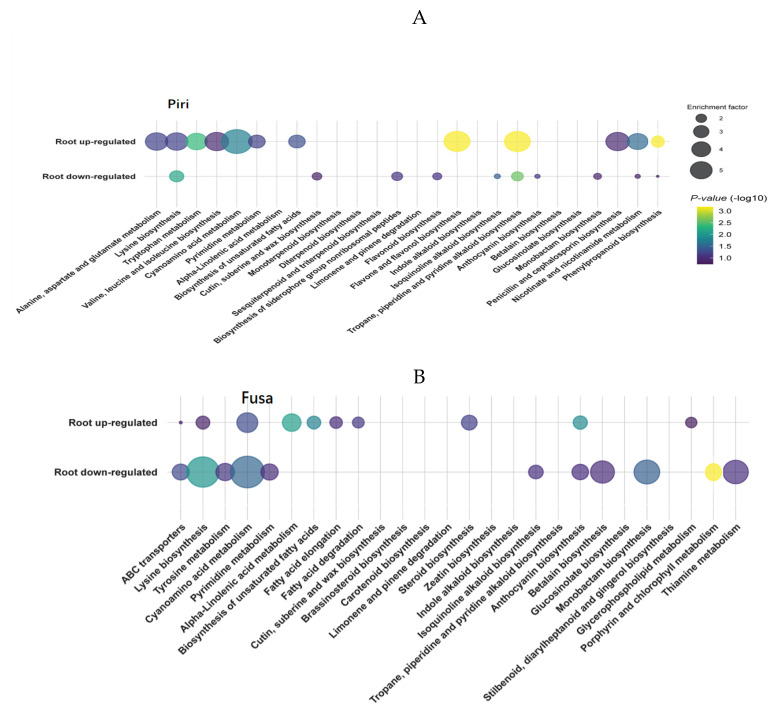
Kyoto Encyclopedia of Genes and Genomes (KEGG) enrichment pathways analysis of altered metabolites under *P. indica and F. pseudograminearum* colonization. (**A**) Enrichment pathways of metabolites following *P. indica* inoculation; (**B**) Enrichment pathways of metabolites following *F. pseudograminearum* infection; (**C**) Enrichment pathways of metabolites following *P. indica* + *F. pseudograminearum* infection. Bubble size represents the number of enrichment factors in the pathway. Bubble color changes from purple to yellow indicates greater statistical significance.

**Figure 10 ijms-24-08797-f010:**
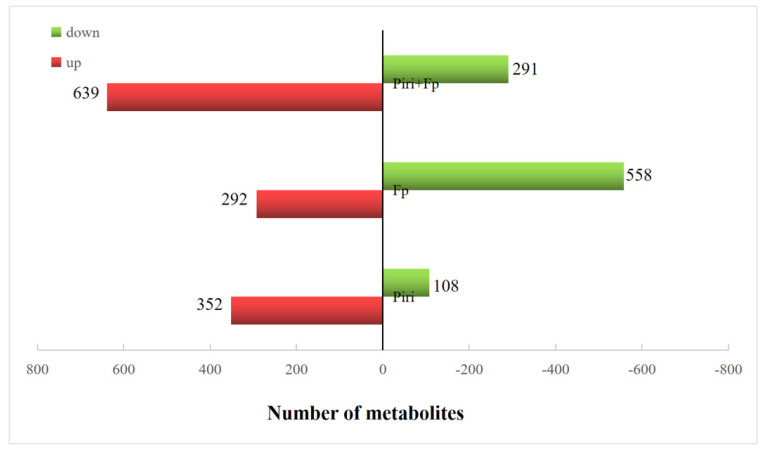
Metabolic changes in wheat roots under *P. indica* colonization in the jointing stage. Differential metabolite accumulation in the roots of wheat under *P. indica, F. pseudograminearum* and *P. indica* + *F. pseudograminearum* colonization. Red columns represent up-regulated metabolites, and green columns represent down-regulated metabolites. Six biological replicates were implemented for each treatment, and each biological replicate comprises a pool of 10 plants.

**Figure 11 ijms-24-08797-f011:**
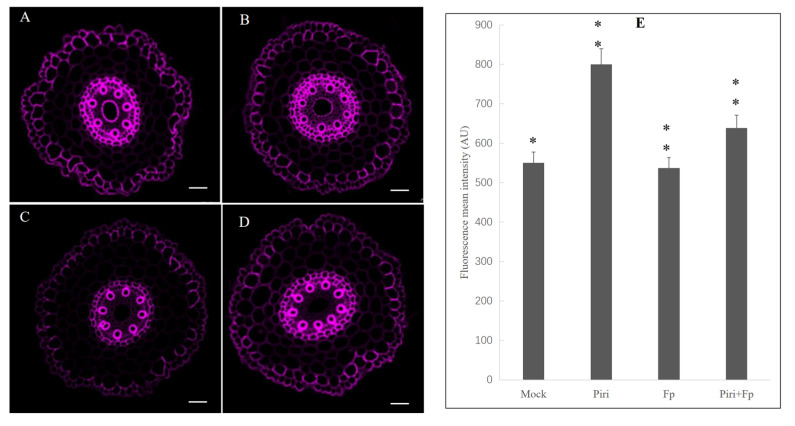
Lignin staining of wheat roots 48 h after inoculation with *P. indica and F. pseudograminearum* spores. For lignin detection root sections were stained with 0.2% (*w*/*v*) Basic Fuchsin in ClearSee solution for overnight. (**A**) Control. (**B**) *P. indica* colonization 48 h; (**C**) *F. pseudograminearum* infection 48 h; (**D**) *P. indica* precolonization 48 h and infected with *F. pseudograminearum* infection 48 h. Note: strongly increased lignin staining in surface cell layers, endodermis, and xylem of both *P. indica* colonization lines and *P. indica* + *F. pseudograminearum* lines. Quantification of lignin staining in freehand wheat root cross-sections. Fluorescence mean intensity is determined in arbitrary units (AU). (**E**) Data are presented as means ± standard deviation (SD) from at least two roots of each wheat line in three biological replicates (*n* = 6). ** above the columns represents significant differences (*p* < 0.01),* above the columns represents significant differences (*p* < 0.05). Scale bars: 50 µm.

**Table 1 ijms-24-08797-t001:** The up-regulated induction of key DEGs involved in lignin synthesis pathway.

ID	NCBI Accession	Description	GO	KEGG Pathway
TraesCS2D02G377200	XM_044477614.1	Triticum aestivum phenylalanine ammonia-lyase-like (PAL)	GO: 0045548 phenylalanine ammonia-lyase activity GO: 0006559 L-phenylalanine catabolic process GO: 0009800 cinnamic acid biosynthetic process	00360 Phenylalanine metabolism 00940 Phenylpropanoid biosynthesis
TraesCS5A02G213900	XM_044525426.1	Triticum aestivum probable cinnamyl alcohol dehydrogenase 8D	GO: 0008270 zinc ion binding GO: 0016616 oxidoreductase activity, acting on the CH-OH group of donors, NAD or NADP as acceptor GO: 0045551 cinnamyl-alcohol dehydrogenase activity GO: 0052747 inapyl alcohol dehydrogenase activity GO: 0009809 lignin biosynthetic process	00940 Phenylpropanoid biosynthesis
TraesCS7A02G084600	XM_044570301.1	Triticum aestivum probable O-methyltransferase 2	GO: 0008171 O-methyltransferase activity GO: 0008757 S-adenosylmethionine-dependent methyltransferase activity GO: 0046983 protein dimerization activity GO: 0019438 aromatic compound biosynthetic process GO: 0032259 methylation	00945 Stilbenoid, diarylheptanoid, and gingerol biosynthesis
TraesCS2B02G395400	XM_044469367.1	Triticum aestivum caffeoyl shikimate esterase-like		00561 Glycerolipid metabolism
(4CL) TraesCS6B02G294100	XM_044557718.1	Triticum aestivum 4-coumarate-oA ligase 2-like	GO: 0016207 4-coumarate-CoA ligase activity GO: 0009698 phenylpropanoid metabolic process	00130 Ubiquinone and other terpenoid-quinone biosynthesis 00940 Phenylpropanoid biosynthesis
TraesCS7D02G239400	XM_044583893.1	Triticum aestivum tricin synthase 2	GO: 0042409 caffeoyl-CoA O-methyltransferase activity CCOACMT GO: 0046872 metal ion binding GO: 0009809 lignin biosynthetic process GO: 0032259 methylation	
TraesCS5B02G268300	XM_044533418.1	Triticum aestivum cinnamoyl-CoA reductase 1-like	GO: 0016491 oxidoreductase activity GO: 0016616 oxidoreductase activity, acting on the CH-OH group of donors, NAD or NADP as acceptor GO: 0016621 cinnamoyl-CoA reductase activity GO: 0050662 coenzyme binding GO: 0007623 circadian rhythm GO: 0009809 lignin biosynthetic process	00940 Phenylpropanoid biosynthesis

**Table 2 ijms-24-08797-t002:** The up-regulated induction of key DEGs involved in flavonoid synthesis pathway.

ID	NCBI Accession	Description	GO	KEGG Pathway
TraesCS5D02G488700	XM_044543986.1	Triticum aestivum chalcone synthase 2-like(CHS)	GO: 0016210 naringenin-chalcone synthase activity GO: 0009813 flavonoid biosynthetic process	00941/Flavonoid biosynthesis 04712 Circadian rhythm—plant
TraesCS5D02G489000	XM_044546410.1	Triticum aestivum chalcone--flavanone isomerase-like	GO: 0045430 chalcone isomerase activity GO: 0009813 flavonoid biosynthetic process (CHI)	00941 Flavonoid biosynthesis
TraesCS7A02G333900	XM_044567041.1	Triticum aestivum flavone O-methyltransferase 1-like	GO: 0008171 O-methyltransferase activity GO: 0008757 S-adenosylmethionine-dependent methyltransferase activity GO: 0046983 protein dimerization activity GO: 0009809 lignin biosynthetic process GO: 0009813 flavonoid biosynthetic GO: 0019438 aromatic compound biosynthetic process GO: 0032259 methylation	
TraesCS2B02G103600	XM_044466025.1	Triticum aestivum isoflavone reductase homolog (IFR)	GO: 0016491 oxidoreductase activity	00998 Biosynthesis of various secondary metabolites—part 2
TraesCS7D02G152300	XM_044587211.1	Triticum aestivum putative anthocyanidin reductase(ANR)	GO: 0003824 catalytic activity GO: 0050662 coenzyme binding	
TraesCS4D02G227300	XM_044519958.1	Triticum aestivum UDP-glycosyltransferase 83A1-like	GO: 0016758 transferase activity, transferring hexosyl groups	

**Table 3 ijms-24-08797-t003:** Respective treatments for different samples.

Number	Treatments	Abbreviation	Repeats	*P. indica* Pre-Inoculation	Harvest at () day
1	*Mock*	Mock	6	-	14^0^ + 14^0^
2	*Piriformospora indica*	Piri	6	+	14^pi^ + 14^0^
3	*F.* *pseudograminearum*	Fp	6	-	14^0^ + 14^fp^
4	*P. indica* + *F. pseudograminearum*	Piri + Fp	6	+	14^pi^ + 14^fp^

Note: 14^0^ stands for days irrigation with water; 14^fp^ stands for *Fusarium graminearum* infection for 14 days; 14^pi^ stands for days pre-inoculation of *P. indica*.

## Data Availability

No data were used for the research described in the article.

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
