# Peer review of "Piriformospora indica Increases Resistance to Fusarium pseudograminearum in Wheat by Inducing Phenylpropanoid Pathway"

_ijms, 2023, doi:10.3390/ijms24108797_

Round 1

Reviewer 1 Report

Comments to the manuscript

Piriformospora indica increases resistance to Fusarium pseudograminearum in wheat by inducing phenylpropanoid pathway 

Liang Li, Ruiying Hao, Xiurong Yang, Yu Feng and Zhenghui Bi

Fusarium pseudograminearum  is a very  serious problem for wheat production worldwide as it not only decreases yield but also produces toxins that are harmful for humans and animals. The use of biological means is an environment-friendly approach of plant protection. Piriformospora indica colonization of roots was shown to alleviate the impact of the Fusarium infection on wheat plants. In this manuscript the authors not only demonstrate the effect of P. indica pretreatment on improvement of plant resistance but more importantly elucidate molecular mechanisms of this process. The results of this manuscript are of importance for understanding mechanisms of plant resistance and provide the basis for bilogical control of Fusarium crown rot.

Some corrections are recommended for the manuscript.

·         Please revise the phrases in lane 88 “The role of phenylpropanoid in wheat colonized by P. indica responses to Fusarium has not been elucidated,..”, in lane 153 “per wheat strain” (there was only one cultivar in the experiment); in lane 155 “Construct an mRNA library using the samples in Table 1”; in lane 165 “Format the single stranded circular DNA (ssCir DNA) into the final library”; in lanes 170, 171 “.. stands for days irrigation with water” (and look through the whole Note); lane 218 “...the differences of significance” (maybe,  the significance of differences..); lane 237 “And  the  pathogen  of F. pseudograminearum led to...”; lane 244  “the seed of pre-colonized with P. indica was less affected..”; lane 448 Monica et al., 2008 (Sharma?)

·         in lane 224 please replace Fig. 1B by Fig.1A.; in lanes 227 and 533 Fig. 1D by Fig. 1C.

·         In Fig1 I found arrows but failed to find circles indicating mycelia of P. indica (lane 234).

·         Please add to lane 123 “Scanning Electron Microscopy (SEM)” instead of SEM, and in lane 225 replace this phrase by SEM

·         In 2.3. Plant Disease Evaluation there are only scores but in in Fig.2 B there is disease index on the y-axis. Please add to this section how the disease index was calculated and in what units.

·         In Fig.2 there is “days post inoculation (dpi)”, but in Table 1 and other parts of the manuscripts there is “dai” (days after inoculation), please replace dpi by dai in Fig.2.

·         The procedure of quantification of F. pseudograminearum colonization by PCR (Fig.3, sec. 3.3) is not described in Materials and Methods. Please mention it in Materials and Methods.

·         In Fig.2 captions, in the phrases “Plant dry weight was calculated..” and “Stem dry weight was calculated” (lanes 253, 253) please replace “calculated” by “determined”.

·         Lane 260: ”The DON toxin content in wheat roots increased with the accumulation of F. pseudograminearum infection time”, but this cannot be seen from bars in Fig.3B, considering SE.

·         In Table 1 and other parts of the manuscript the abbreviation Piri+Fp is used, so please use this abbreviation also in Fig.4.

·         It seems likely that in lane 300 the number of down-regulated DEGs should be 17.828 and in lane 301 the number of up-regulated DEGs – 13.527.

·         Similarly, please change reciprocally the numbers of down- and up-regulated DEGs in lane 304.

·         In the manuscript there are no tables 2 and 3 mentioned in lane 341.

·         Please add to captions for Fig. 7 and 8, respective letters A-F for each enzyme.

·         Fig.10 is not mentioned in the text!

·         In Fig.11 “Different small letters above the columns represent significant differences (P<0.05).” Please check the letters and significance of differences for the results in Fig.11 and other figures with bars, as, for example in Fig.11 the variant Piri does differ from Mock but they both have the same letter “a”.

Author Response

Thank you very much for your handling of our manuscript and for your positive comments. We have responded to the comments raised by the reviewer as outlined below and would now like to submit the revised manuscript for consideration of publication in IJMS. We thank you and the reviewers for comments and suggestions that are extremely valuable for improving our manuscript.

“Some corrections are recommended for the manuscript.

  • Please revise the phrases in lane 88 “The role of phenylpropanoid in wheat colonized by indicaresponses to Fusarium has not been elucidated,..”,
  • in lane 153 “per wheat strain” (there was only one cultivar in the experiment);
  • in lane 155 “Construct anmRNA library using the samples in Table 1”;
  • in lane 165 “Format the single stranded circular DNA (ssCir DNA) into the final library”;
  • in lanes 170, 171 “.. stands for days irrigation with water” (and look through the whole Note);
  • lane 218 “...the differences of significance” (maybe,  the significance of differences..);
  • lane 237 “And  the  pathogen  of F. pseudograminearum led to...”;
  • lane 244  “the seed of pre-colonized with P. indica was less affected..”;
  • lane 448 Monica et al., 2008 (Sharma?)
  • in lane 224 please replace Fig. 1B by Fig.1A.; in lanes 227 and 533 Fig. 1Dby Fig. 1C.
  • In Fig1 I found arrows but failed to find circles indicating mycelia of P. indica(lane 234).
  • Please add to lane 123 “Scanning Electron Microscopy (SEM)” instead of SEM, and in lane 225 replace this phrase by SEM
  • In 2.3. Plant Disease Evaluation there are only scores but in in Fig.2 B there is disease index on the y-axis. Please add to this section how the disease index was calculated and in whatunits.
  • In Fig.2 there is “days post inoculation (dpi)”, but in Table 1 and other parts of the manuscripts there is “dai” (days after inoculation), please replace dpiby dai in Fig.2.
  • The procedure of quantification of F. pseudograminearum colonizationby PCR (Fig.3, sec. 3.3) is not described in Materials and Methods. Please mention it in Materials and Methods.
  • In Fig.2 captions, in the phrases “Plant dry weight was calculated..” and “Stem dry weight was calculated” (lanes 253, 253) please replace “calculated” by “determined”.
  • Lane 260: ”The DON toxin content in wheat roots increased with the accumulation of F. pseudograminearum infection time”, but this cannot be seen from bars in Fig.3B, considering SE.
  • In Table 1 and other parts of the manuscript the abbreviation Piri+Fpis used, so please use this abbreviation also in Fig.4.
  • It seems likely that in lane 300 the number of down-regulated DEGs should be 17.828 and in lane 301 the number of up-regulated DEGs – 13.527.
  • Similarly, please change reciprocally the numbers of down- and up-regulated DEGs in lane 304.
  • In the manuscript there are no tables 2 and 3mentioned in lane 341.
  • Please add to captions for Fig. 7 and 8, respective letters A-Ffor each enzyme.
  • 10 is not mentioned in the text!
  • In Fig.11 “Different small letters above the columns represent significant differences (P<0.05).” Please check the letters and significance of differencesfor the results in Fig.11 and other figures with bars, as, for example in Fig.11 the variant Piri does differ from Mock but they both have the same letter “a”.”

The all comments raised by the first reviewer were followed and corrected, please find them in the MS.

Reviewer 2 Report

The role and potential of Piriformospora indica for improving wheat resistance against diseases and particularly against Fusarium crown rot is a relatively new and not studied enough field of knowledge. The manuscript makes a new contribution into the better understanding of interaction between wheat and Piriformospora indica and possible mechanism of disease resistance to FCR.

The paper is well-written and supported by appropriate literature. However, there is still a need to make certain corrections. All my comments and suggestions are in the text of the manuscript.

Author Response

Thank you very much for your handling of our manuscript  and for your positive comments. We have responded to the comments raised by the reviewer as outlined below and would now like to submit the revised manuscript for consideration of publication in IJMS. We thank you and the reviewers for comments and suggestions that are extremely valuable for improving our manuscript.Some corrections are recommended for the manuscript.

Reviewer 2

1.Could you add the photo of pre-colonizated plants with P. indica alone, without F. pseudograminearum to demonstrate this phenomenon visually. Was it difficult to differentiate stem browning caused by F. pseudograminearum and P. indica?

We add a new photo already!

2.Which values are presented in the Figure 2B, the average values of five evaluations? If disease severity was evaluated several times with the same interval, it would be great to demonstrate the area under the disease progress curve (AUDPC).

 We added this part int he M&M.

3."each biological replicate comprises a pool of 10 plants" it is not mentioned in Materials and Methods. Did you use 10 plants as one replication across all experiments or only for metabolic analysis?

We  used 10 plants as one replicationonly for metabolic analysis and repeated 6 times, this part has been added to the M&M  now.

4.P. indica cause 27216 DEGs (Mock vs Piri_Fp) 31355 DEGs was in case Mock vs Fp.

In figure 4B, the DEGs in Mock vs Piri was 641 indeed, that is why we mentioned that P. indica did not cause much differential gene expression.

And all in all, the  comments raised by the first reviewer were followed and corrected, please find them in the MS.
